# First Report of Azole-Resistant *Aspergillus fumigatus* with TR_46_/Y121F/T289A Mutations in Kuwait and an Update on Their Occurrence in the Middle East

**DOI:** 10.3390/jof9080784

**Published:** 2023-07-25

**Authors:** Mohammad Asadzadeh, Khaled Alobaid, Suhail Ahmad, Sara Mazloum

**Affiliations:** 1Department of Microbiology, College of Medicine, Kuwait University, Safat 13110, Kuwait; mohammad.assadzadeh@ku.edu.kw; 2Mycology Reference Laboratory, Mubarak Al-Kabeer Hospital, Ministry of Health, Jabriya 46300, Kuwait; khaled22m@live.com; 3Microbiology Laboratory, Jaber Al-Ahmad Hospital, Ministry of Health, South Surra 91711, Kuwait; sara7.ma7moud@yahoo.com

**Keywords:** *Aspergillus fumigatus*, voriconazole resistance, Cyp51A, TR_46_/Y121F/T289A mutation, Middle East

## Abstract

Pulmonary aspergillosis is a common fungal infection with several clinical manifestations including invasive, allergic and chronic chest diseases. Invasive pulmonary aspergillosis (IPA) is a leading cause of death in immunocompromised patients, particularly those receiving chemotherapy and among bone marrow transplant recipients. *Aspergillus fumigatus* is the most prevalent causative agent and voriconazole is the first-line therapy for IPA. In this study, we report the first isolation of voriconazole-resistant *A. fumigatus* carrying TR_46_/Y121F/T289A mutations from an immunocompromised pregnant lady in Kuwait. The patient was successfully treated for a probable respiratory infection with caspofungin and voriconazole. The literature review from PubMed has identified itraconazole-resistant clinical and environmental *A. fumigatus* isolates with TR_34_/L98H mutations in the *cyp51A* from several Middle Eastern countries including Kuwait. However, clinical *A. fumigatus* isolates with *cyp51A* TR_46_/Y121F/T289A mutations have not been reported previously from any country in the region while environmental isolates have been reported only from Iran. The source of voriconazole-resistant *A. fumigatus CYP51A* TR_46_/Y121F/T289A mutant in our patient remained unknown. Surveillance for azole resistance among clinical and environmental isolates of *A. fumigatus* is warranted in Kuwait.

## 1. Background

Invasive fungal infections (IFIs) are mostly caused by opportunistic pathogens in nosocomial settings [1,2,3]. Opportunistic yeast and mold infections are increasing worldwide among hospitalized patients due to the increasing use of antifungal prophylaxis in an expanding population of immunocompromised patients and other susceptible patients with multiple comorbidities [4,5,6,7]. The epidemiology of IFIs is also changing rapidly due to the increasing incidence of antifungal drug resistance in commonly encountered yeasts and molds [7,8,9,10,11,12,13]. The emergence of novel fungal pathogens with reduced susceptibility or resistance to one or more antifungal drugs, with some species (e.g., *Candida auris*) causing major outbreaks in healthcare facilities in many countries, is another major concern [3,14,15,16]. The IFIs need prompt detection and precise antifungal susceptibility testing of the causative agents to common antifungals for proper patient management, as these infections are usually associated with high mortality rates in critically ill patients [17,18,19,20].

*Aspergillus fumigatus* (AF) is a widespread saprophytic fungus that causes several clinical manifestations, including invasive, allergic, and chronic chest diseases. Invasive pulmonary aspergillosis (IPA) has become a major concern among immunocompromised patients, including solid organ and hematopoietic stem cell transplant recipients as well as those receiving highly immunosuppressive chemotherapies [21]. The azole class of drugs has become a hallmark in the treatment of IPA since they are highly active against *Aspergillus* species; itraconazole, posaconazole, voriconazole, and isavuconazole can be orally or parenterally administered, and the therapy regimen/duration can be altered for IPA refractory to voriconazole [22,23,24,25]. However, prolonged therapy also predisposes patients to adverse drug effects and interactions and increases the possibility of the development of drug-resistant organisms [25,26].

Mutations in the *cyp51A* gene, which encodes a 14-α-sterol demethylase participating in ergosterol production, have been identified as the major mechanism conferring resistance to triazoles in *A. fumigatus* [7,23]. Over the last several years, concern has been growing regarding the emergence of azole resistance in *A. fumigatus* due to agricultural use [25,27]. Azole-resistant *A. fumigatus* isolates harboring nonsynonymous mutations in *cyp51A* have been recovered from patients with prolonged azole exposure and failing therapy [22,23,24]. Resistance-conferring mutations comprising tandem repeats (TR) in the promotor region and nonsynonymous mutation(s) in *cyp51A* linked primarily to environmental use of azoles in agriculture have also been detected in *A. fumigatus* isolates from azole-naïve patients [27,28,29,30]. Cyp51A mutations generally affect the susceptibility of *A. fumigatus* to all triazoles; however, the TR_34_/L98H mutation confers high-level resistance to itraconazole, while isolates with the TR_46_/Y121F/T289A mutation are highly resistant to voriconazole [7,27,30,31]. The TR_34_/L98H mutation in *Cyp51A* in *A. fumigatus* is distributed worldwide, while the TR_46_/Y121F/T289A mutation is an emerging mechanism detected mostly in some European and Asian countries [27,30]. Although >130 clinical and environmental *A. fumigatus* isolates collected from 2008 to 2012 were screened, cyp51A TR_46_/Y121F/T289A mutants were not detected in our previous studies from Kuwait [28,31,32]. Here, we describe the first isolation of *A. fumigatus* with the TR_46_/Y121F/T289A mutation from a treatment-naïve immunocompromised pregnant lady in Kuwait and highlight the emerging role and geographic distribution of this novel mutation in Middle Eastern countries.

## 2. Case Report

A 33-year-old immunocompromised pregnant lady was admitted to our hospital with a fever and watery diarrhea (Day 1). She has previously been diagnosed with Crohn’s disease and was being treated with infliximab 10 mg and prednisolone 5 mg. On admission, she required inotropic support due to vital instability. A computed tomography (CT) of the abdomen revealed colitis, for which the patient received hydrocortisone, piperacillin/tazobactam, and metronidazole. During the next three days, her clinical condition improved, she delivered a baby by cesarean section on Day 4 and was moved to the ward on Day 5. On Day 7, the patient developed tachycardia and hypotension and was readmitted to the ICU. Empiric treatment with vancomycin and caspofungin was commenced because of sepsis. Her chest CT scan showed pleural effusions and collapsed lungs. On Day 11, a pigtail was inserted, and oxygen treatment was administered by nasal cannula. Subsequently, she underwent a bronchoscopy, which showed a large mucus plug in her left bronchial tree. A bronchoalveolar lavage (BAL) sample was collected and sent to the microbiology laboratory, where a mold was grown in culture and was subsequently identified as *A. fumigatus*. Based on initial microbiological findings, caspofungin was replaced by voriconazole (340 mg loading dose BD, followed by 240 mg BD) on Day 18. She remained in the ICU for an additional three days while maintaining adequate oxygen saturation in ambient air. One week later, the infectious disease physician discontinued voriconazole. The patient stayed for two additional weeks in the hospital due to a pelvic hematoma that was surgically removed and was discharged on Day 36.

### Methods

The BAL culture yielded a mold that was initially identified as an *Aspergillus* species based on culture characteristics on Sabouraud dextrose agar and microscopic appearance with lactophenol cotton blue stain. The antifungal susceptibility testing of the clinical isolate was performed using Etest (bioMérieux) following the manufacturer’s instructions and as reported previously [33]. The minimum inhibitory concentration (MIC) values were obtained after 24 h. The *A. fumigatus* isolate (Kw145-8/22) appeared susceptible to amphotericin B (MIC of 0.094 µg/mL), itraconazole (MIC of 0.38 µg/mL), posaconazole (MIC of 0.125 µg/mL), and caspofungin (MIC of 0.032 µg/mL) but resistant to voriconazole (MIC of 16 µg/mL).

Species-specific identification of the isolate was performed using PCR sequencing of β-tubulin gene fragment. Genomic DNA from the isolate was obtained by using the DNeasy Plant Mini Kit (QIAGEN, Hilden, Germany) and following the manufacturer’s instructions. The variable region of β-tubulin gene was amplified by using BTUBF (5′-TGGTAACCAAATCGGTGCTGCTT-3′) and BTUBR (5′-GCACCCTCAGTGTAGTGACCCT-3′) primers and sequenced as described previously [33]. The β-tubulin gene sequence showed differences at only two nucleotide positions with the corresponding sequence from the *A. fumigatus* reference strain ATCC 204305 [34].

The *Cyp51A* gene, including the 5’-untranslated and 3′-untranslated regions, was amplified and sequenced as three overlapping fragments by using the PCR amplification and sequencing protocols described previously [28]. The 5′-region, middle region, and 3′-region were amplified and sequenced by using AFCYP51F1 (5′-CAGCGGCAGCATTCTGAAACA-3′) + AFCYP51R1 (5′-CCGCATTGACATCCTTGAG-3′), AFCYP51F2 (5′-AGTTCTTCTTTGCGTGCAGA-3′) + AFCYP51R2 (5′-GTTCCATATCATGTCTGATTTCT-3′) and AFCYP51F1 (5′-ATGAGGTCAATCTACGTTGA-3′) + AFCYP51R1 (5′-CGAGGGGCTGAATTGTATAA-3′) primers, respectively. The sequence of the entire *Cyp51A* gene was assembled and compared with the wild-type sequence from *A. fumigatus* Af293 used as a reference [34] (NCBI Reference Sequence: NC_007197.1). Our data showed that isolate Kw145-8/22 contained a tandem repeat of 46 nucleotides (TR_46_) in the promoter region and two nonsynonymous mutations, Y121F and T289A, within the coding region. The DNA sequence data for partial β-tubulin and *Cyp51A* genes for isolate Kw145-8/22 have been submitted to GenBank under accession numbers OQ789927 and OQ789928.

## 3. Comments

*Aspergillus* and other mold species are common etiological agents of respiratory infections in susceptible patients [35,36,37,38]. Azole-resistant A. *fumigatus* is widespread in the environment and has been isolated from both outdoor and indoor environments, including tertiary care centers [7,22,23,25,39]. Our study described the isolation of voriconazole-resistant *A. fumigatus* carrying TR_46_/Y121F/T289A mutations in *cyp51A* for the first time from a treatment-naïve patient in Kuwait. The patient was a 33-year-old immunocompromised pregnant lady with Crohn’s disease and was treated with caspofungin followed by voriconazole. The records of her trips abroad were not available, and the environmental sampling for the isolation of similar isolates from the hospital premises and outside was not performed. Although the source of this isolate is not known, it is unlikely that the resistance evolved in the patient, considering there was no previous history of treatment with azole antifungals. Although the isolate was subsequently found to be voriconazole-resistant in vitro, it is probable that the initial course of treatment with caspofungin was sufficient to clear the infection. Voriconazole has also been used either as the initial or modified therapy for successful treatment of patients with *A. fumigatus* carrying TR_46_/Y121F/T289A mutations in *cyp51A* [40].

Previous studies carried out nearly a decade ago have described the isolation of itraconazole-resistant *A. fumigatus* carrying TR34/L98H mutations in *cyp51A* from the hospital’s indoor and outdoor environments [28,31,32]. Although similar isolates were also obtained from the soil and other environmental samples across Kuwait as well as from clinical specimens from the patients, their origin due to agricultural use of fungicidal azoles within Kuwait or their arrival due to aerial dispersion from other nearby countries remained unknown [28,31,32]. Since no environmental screening has been performed for nearly 10 years after our previous studies, it is probable that voriconazole-resistant *A. fumigatus* carrying TR46/Y121F/T289A mutations in *cyp51A* has also now become prevalent within the environment in Kuwait. It is, therefore, reasonable to assume that the patient most likely acquired voriconazole-resistant *A. fumigatus cyp51A* TR46/Y121F/T289A mutants from the environment in Kuwait. However, the possibility that the patient was infected with a mixture of voriconazole-resistant and -susceptible strains simultaneously could not be ruled out since the BAL sample was cultured only once and the antifungal susceptibility testing was carried out on a single colony. It is also probable that the patient recovered after voriconazole treatment as the susceptible strain was removed, which was responsible for larger growth in the lung. Simultaneous presence of triazole-resistant and -susceptible strains of *A. fumigatus* in clinical and environmental samples has been described previously [32,39,41,42].

IPA is a life-threatening fungal infection associated with high morbidity, mortality, and prolonged hospital stays, particularly among severely immunocompromised patients due to cancer treatment, organ transplantation, or prolonged immunosuppressive therapy [21,24,39]. Isolation of triazole-resistant *A. fumigatus*, particularly the *A. fumigatus cyp51A* TR_46_/Y121F/T289A mutant, is a worrisome development since voriconazole is the first-line therapy for IPA and invasive disease with this genotype is associated with therapy failure [41]. Isolation of triazole-resistant *A. fumigatus* with TR_34_/L98H and TR_46_/Y121F/T289A mutations from environmental sources and treatment-naïve patients in several European and Asian countries has been attributed to the widespread use of fungicidal azoles in agriculture in these countries [27,39]. Isolation of itraconazole-resistant *A. fumigatus* with TR_34_/L98H mutations in outdoor and hospital environments as well as from clinical specimens from patients had previously extended the distribution of this genotype in the Arabian Gulf region [28,31,32]. *A. fumigatus* with TR_34_/L98H mutations have also been previously reported from Iran and Turkey [43,44,45,46,47,48,49,50] (Table 1).

To the best of our knowledge, this is the first report of an *A. fumigatus cyp51A* TR_46_/Y121F/T289A mutant isolated from a clinical specimen in Kuwait and the Middle East. So far, only one environmental isolate of *A. fumigatus cyp51A* TR_46_/Y121F/T289A mutant has been reported from Iran, another country in the Middle East [47]. This study from Iran also reported the isolation of another *A. fumigatus* strain with the *cyp51A* TR_46_/Y121F/M172I/T289A/G448S mutation [47]. Other countries (Qatar and Lebanon) in the region have not reported the presence of the TR_34_/L98H mutation or this (*cyp51A* TR_46_/Y121F/T289A mutant) genotype among environmental or clinical isolates of *A. fumigatus* so far [51,52] (Table 1). Our findings advocate the need for active surveillance for antifungal resistance in *Aspergillus fumigatus* and other common molds similar to yeasts to explore the real resistance rates [53]. Surveillance information may also be used to inform decisions regarding health services and research funding allocation, to guide local infection control in hospitals and communities, and to direct local and national drug policies and guidelines [41,53].

## 4. Conclusions

In conclusion, a voriconazole-resistant *A. fumigatus cyp51A* TR_46_/Y121F/T289A mutant mainly linked with fungicidal azole usage has been isolated for the first time from a treatment-naïve patient in Kuwait. The source of the infection remained unknown. The literature search has identified only one previous report of *A. fumigatus* with TR_46_/Y121F/T289A mutations in *cyp51A* from the environment in Iran among nearby countries. Environmental screening is clearly warranted to determine the prevalence of triazole-resistant *A. fumigatus* in the air/soil samples within Kuwait.

## Figures and Tables

**Table 1 jof-09-00784-t001:** Distribution of triazole-resistant *A. fumigatus* in clinical and environmental samples in the Middle East containing mutations in the *Cyp51A* gene.

No.	Country	Year	Source	No. of Screened Isolates/Samples	No. of Confirmed AF	No. (%) of Triazole Resistant	No. (%) of Isolates with TR34/L98H Mutations	No. (%) of Isolates with TR46/Y121F/T289A Mutations	Reference
1	Iran	2013	Clinical	124	124	4 (3.2)	3 (75)	0	[43]
2	Iran	2013	Environment	150	41	5 (12.2)	5 (100)	0	[44]
3	Iran	2016	Clinical	213	71	3 (4.2)	2 (66.7)	0	[45]
			Environment	300	79	7 (8.9)	6 (85.7)	0	
4	Iran	2018	Environment	108	31	4 (13)	2 (50)	0	[46]
5	Iran	2020	Environment	300	63	57 (90.5)	44 (77)	2 (3.5) *	[47]
6	Kuwait	2014	Environment	362	115	8 (7)	8 (100)	0	[31]
7	Kuwait	2015	Clinical	16	16	1 (6.3)	1 (6)	0	[32]
			Environment	50	50	0	0	0	
8	Kuwait	2022	Clinical	1	1	1 (100)	0	1	Current study
9	Turkey	2015	Clinical	746	746	76 (10.2)	66 (86.8)	0	[48]
10	Turkey	2018	Clinical	31	31	1 (3.2)	0	0	[49]
11	Turkey	2022	Clinical	392	392	13 (3.3)	9 (69)	0	[50]
			Environment	2288	457	6 (1.3)	0	0	
12	Qatar	2019	Clinical	70	12	1 (8.3)	ND	ND	[51]
13	Lebanon	2022	Environment	262	28	1 (3.6)	0	0	[52]

* One isolate harboring TR_46_/Y121F/M172I/T289A/G448S mutations. AF = *Aspergillus fumigatus*; ND = Not determined.

## Data Availability

All relevant data are available in the manuscript.

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
