# Peer review of "First Report of Azole-Resistant Aspergillus fumigatus with TR46/Y121F/T289A Mutations in Kuwait and an Update on Their Occurrence in the Middle East"

_jof, 2023, doi:10.3390/jof9080784_

Round 1

Reviewer 1 Report

the paper describes the identification of a voriconazole-resistant A. fumigatus cyp51A TR46/Y121F/T289A mutant found for the first time in Kuwait. The manuscript is clearly written. The novelty is solely in the fact that this isolate is described for the first time in Kuwait. 

the isolate was characterized after isolation from a BAL. The patient did not receive voriconazole at that moment so it is reasonable to belive it had an environmental origin. Question is if the authors characterized multiple A fum isolates from the BAL. this is not described. It is not uncommon to have multiple isolates from one patient and which can be mixtures of inhaled spores and mycelia. The patient could have carried resistant and non-resistant strains. Possibly the patient could also have recovered after voriconazole treatment as the sensitive strains was removed and which was responsible for larger growth in the lung. The authors should discuss this issue as well. Now, they describe the story as if the patient carried only one fungal isolate that was resistant.

ref 30 is now numbered as 3

Author Response

Reviewers comments:
the paper describes the identification of a voriconazole-resistant A. fumigatus cyp51A TR46/Y121F/T289A mutant found for the first time in Kuwait. The manuscript is clearly written. The novelty is solely in the fact that this isolate is described for the first time in Kuwait.

Authors response: We thank the reviewer for the positive comments. No specific comments to respond to here.

Reviewers comments:
the isolate was characterized after isolation from a BAL. The patient did not receive voriconazole at that moment so it is reasonable to belive it had an environmental origin. Question is if the authors characterized multiple A fum isolates from the BAL. this is not described. It is not uncommon to have multiple isolates from one patient and which can be mixtures of inhaled spores and mycelia. The patient could have carried resistant and non-resistant strains. Possibly the patient could also have recovered after voriconazole treatment as the sensitive strains was removed and which was responsible for larger growth in the lung. The authors should discuss this issue as well. Now, they describe the story as if the patient carried only one fungal isolate that was resistant.

Authors response: We thank the reviewer for these comments. This point has now been elaborated in the revised manuscript as suggested by the reviewer.

Reviewers comments:
ref 30 is now numbered as 3

Authors response: We thank the reviewer for pointing out this error which has now been corrected.

Reviewer 2 Report

This article is the description of a case study of a patient infected with A. fumigatus. The description of the case study is well down. I would introduce a methods section at line 96 to differentiate between the actual case and the investigation performed.

No comments on this

Author Response

Reviewers comments:
This article is the description of a case study of a patient infected with A. fumigatus. The description of the case study is well down. I would introduce a methods section at line 96 to differentiate between the actual case and the investigation performed.

Authors response: We thank the reviewer for the positive comments. A sub-heading of the ‘Methods’ has been introduced as suggested by the reviewer.

Reviewer 3 Report

Dear authors:

I consider the article is well presented and it is important to recommend surveillance for azole resintance in your country, as you did.

Author Response

Reviewers comments:

I consider the article is well presented and it is important to recommend surveillance for azole resintance in your country, as you did.

Authors response: We thank the reviewer for the positive comments. No specific comments to respond to.